# Atmospheric blocking induced by the strengthened Siberian High led to drying in west Asia during the 4.2 ka BP event – a hypothesis

Aurel Perşoiu[1,2], Monica Ionita[3], Harvey Weiss[4]

[1]Emil Racoviţă Institute of Speleology, Romanian Academy, Cluj Napoca, 400006, Romania
[2]Stable Isotope Laboratory, Ştefan cel Mare University, Suceava, 720229, Romania
[3]Alfred Wegener Institute, Helmholtz Center for Polar and Marine Research, Bremerhaven, 27570, Germany
[4]School of Forestry and Environmental Studies, Yale University, New Haven, USA

*Correspondence to*: Aurel Perşoiu (aurel.persoiu@gmail.com)

**Abstract.**

Causal explanations for the 4.2 ka BP event are based on the amalgamation of seasonal and annual records of climate variability manifest across global regions dominated by different climatic regimes. However, instrumental and paleoclimate data indicate that seasonal climate variability is not always sequential in some regions. The present study investigates the spatial manifestation of the 4.2 ka BP event during the boreal winter season in Eurasia, where climate variability is a function of the spatio-temporal dynamics of the westerly winds. We present a multi-proxy reconstruction of winter climate conditions in Europe, west Asia and northern Africa between 4.3 and 3.8 ka BP. Our results show that, while winter temperatures were cold throughout the region, precipitation amounts had a heterogeneous distribution, with regionally significant low values in W Asia, SE and N Europe and local high values in the N Balkan Peninsula, the Carpathian Mountains, and E and NE Europe. Further, strong northerly winds were dominating in the Middle East, and E and NE Europe. Analyzing the relationships between these climatic conditions, we hypothesize that in the extratropical Northern Hemisphere, the 4.2 ka BP event was caused by the strengthening and expansion of the Siberian High, which effectively blocked the moisture-carrying westerlies from reaching W Asia, and enhanced outbreaks of cold and dry winds in that region. The behavior of the winter and summer monsoons suggests that when parts of Asia and Europe were experiencing winter droughts, SE Asia was experiencing similar summer droughts, resulting from failed and/or reduced monsoons. Thus, while in the extratropical regions of Eurasia the 4.2 ka BP event was a century-scale winter phenomenon, in the monsoon-dominated regions it may have been a feature of summer climate conditions.

## 1 Introduction

The 4.2 ka BP climate event was a ca. two–three hundred year period of synchronous abrupt megadrought, cold temperatures and windiness manifest globally (Walker et al., 2018). Coincident societal collapses and habitat tracking, particularly in regions where archaeological data are both extensive and high–resolution, have attracted the attention of many paleoclimatologists and archaeologists since the event's first observation (Gasse and van Campo, 1994; Weiss et al., 1993;

Dalfes et al., 1997). Numerous attempts, therefore, have been made to characterize and quantify the event's nature and to
identify its causes at several levels of explanation. These studies have first defined the spatial extent and variability of the
event. Megadrought developed abruptly at ca. 4.2 ka cal BP across North America, Andean South America, the
Mediterranean basin from Spain to Turkey (except for a few records from N Morocco and S Spain which indicate wetter
conditions), Iran, India, Tibet, and north China and Australia (Booth et al., 2005; Staubwasser and Weiss, 2006; Arz et al.,
2006; Berkelhammer et al., 2013; Cheng et al., 2015; Weiss, 2016; Kathayat et al., 2018). In South Asia, failure of the
monsoon (Wang et al., 2005) caused widespread droughts (Staubwasser et al., 2003; Berkelhammer et al., 2013). Abrupt
cold conditions, however, appeared at ca. 4.2 ka cal BP in the north Atlantic (Geirsdottir et al., 2019), the mid–latitudes of
the northern Eurasia (Hughes et al., 2000; Mayewski et al., 2004; Andresen and Björck, 2005; Mischke and Zhang, 2010;
Larsen et al., 2012; Baker et al., 2017), and Antarctica (Peck et al., 2015) and surrounding oceans (Moros et al., 2009).
These descriptive data have encouraged numerous causal hypotheses both at regional and, to a lesser extent, global level
for the event's spatio–temporal distribution and qualities. Possible thermohaline circulation weakening or shutdown due to
freshwater release in the North Atlantic (similar to the 8.2 ka BP event (Alley et al., 1997)), changes in the loading the
Earth's atmosphere with aerosols or $CO_2$ (Walker et al., 2012) and volcanic forcing (Kobashi et al., 2017) have been rejected
as causes (Walker et al., 2012). At regional explanatory levels, cooling of the southern oceans (Moros et al., 2009) could
have resulted in stronger and more frequent El Niño events that would have weakened (or lead to the failure of) the South
Asian monsoons (Morill et al., 2003; Walker et al., 2012).
The abrupt century–scale wet event recorded at very high resolution in North America, at Mt Logan, Yukon (Fisher et al.,
2008) suggests an interval of massive advection of tropical air to NW North America linked to El Niño emergence at ca. 4.2
ka BP (Shulmeister and Lees, 1995). A southward shift of the Inter Tropical Convergence Zone (ITCZ) could result in the
observed cooling at high latitudes and stronger westerlies in the Northern Hemisphere and widespread drought in the tropics
(Gasse and Van Campo, 1994; Mayewski et al., 2004)). However, the widespread droughts both at the northern and southern
margins of the ITCZ suggest that rather than migrating, the ITCZ was narrowing, resulting in megadrought affecting the
tropics both south and north of the Equator (Weiss, 2016). Combining the above observations, it results that while some of
the climate variability at ca. 4.2 ka cal BP can be attributed to regionally observable causes, explanations do not yet account
for the global nature of the event, that is, disruption of the westerlies and reduction of moisture advection to continents.
Hypothesized causal explanations for the 4.2 ka BP event are based on the amalgamation of winter, summer and annual
records of climate variability manifest in regions dominated by different climatic regimes (e.g., westerly dominated *vs.*
monsoon dominated). However, both instrumental (Balling et al., 1998) and paleoclimate data (Perșoiu et al., 2017) indicate
that, on scales ranging from annual to millennial, seasonal climate variability was not always sequential, i.e., warm (cold)
summers were not always followed by warm (cold) winters. To address this conundrum, we have investigated the spatial
manifestation of the 4.2 ka BP event during winter in a region dominated by climate variability induced by the strength and
dynamics of westerly winds. We present a reconstruction of winter climate conditions in Europe, the Near East and northern
Africa, between 4.3 and 3.8 ka cal BP. From examination of the spatial distribution of temperature and precipitation

excursions during this period, we hypothesize that, in the regions around the Eurasian landmass, the 4.2 ka BP event was caused by strengthening and expansion of the Siberian High pressure cell centered over western Asia that caused widespread cooling at mid–latitudes in the Northern Hemisphere and arridification in the Middle East. We further discuss the possible causes and mechanisms leading to this phenomenon in a global perspective.

## 2 Methods

For our analysis, we have selected proxy records from Europe, the Middle East, northern Africa and the Atlantic Ocean that cumulatively fulfilled a set of five criteria on interpretation, chronology, resolution and nature of climatic variability. We have selected only records of winter climate variability, either precipitation amount (the vast majority) or air temperature, as indicated by the authors. Where no season was indicated we assumed that the proxy is recording annual climatic changes and we excluded it from our analysis. We have selected records with at least two absolute age determinations for the millennium encompassing the 4.2 ka BP event and for which measurement uncertainties were less than 50 years. A few high–resolution records from the fringes of the core study area (mainland continental Europe, the Middle East and the Mediterranean Basin) with age uncertainties up to 80 years were nevertheless used to refine the spatial interpretation of the results. To allow for chronological uncertainties, we have selected records that showed the onset of the local event within ±100 years of the accepted onset of the 4.2 ka BP event (Walker et al., 2018) and duration between 50 and 300 years. Further, we have considered only those records that showed both an abrupt onset and termination (arbitrary set to 15 % against the preceding 100 years), matching the widely distributed 4.2 ka BP event onset, and for which at least 5 data points exists for the 4,300–3800 BP interval.

The response of European temperatures and precipitation to the variability of the Siberian High (SH) (Fig. 1) is based on the Climatic Research Unit Timeseries (CRU TS) 4.01 dataset (Harris et al., 2014). The relationship between the SH intensity, Sea Level Pressure (SLP), and 10 m wind has been analyzed within composite maps for the years when the SH index was greater (HIGH) and lower (LOW) than a value of one standard deviation. We have computed composite maps, instead of correlation maps, because the former considers the nonlinearities included in the analyzed data. The SH index has been obtained by averaging the SLP over the key regions between 40° N and 65° N and 80° E and 120° E (Panagiotopoulos et al., 2005). The SLP and 10 m zonal and meridional wind data were extracted from the ERA 20C dataset (Poli et al., 2016). Our analysis has shown that the results are not sensitive to the exact threshold value used for our composite analysis (i.e., varying the standard deviation between 0.5 and 1.5). To isolate the interannual variations, the linear trend has been removed prior to the analysis from the SH index as well as from the analyzed fields.

## 3 Results and discussions

The list of records with information on type of proxy used and its climatic interpretation, chronology and resolution information is presented in Table 1 and plotted in Fig. 1. Of the 30 selected proxies, 11 register winter (or cold season) temperature, and 19 register winter precipitation amount. The temperature sensitive proxies are from central and northern Europe and SW Asia, while the precipitation sensitive proxies cover the entire study area (between 30° W and 80° E, and 20° N and 78° N), with a concentration in Europe, the Middle East and northern Africa (Fig. 1). Both temperature and precipitation sensitive proxies were plotted against the map depicting the correlation between winter (December–January–February, DJF) climate (temperature and precipitation) and a stronger than usual Siberian High (Fig. 1).

### 3.1 Cold Europe and southwest Asia

The 4.2 ka BP event appears generally as cold during winter throughout Europe, from the Urals to the Atlantic Ocean (Fig. 1a). The highest amplitude of cooling is seen in the Ural Mountains (Baker et al., 2017), at high altitude in the Alps (Fohlmeister et al., 2013), both recorded by speleothem $\delta^{18}$O; and in Central Asia (Wolff et al., 2017) recorded by speleothem $\delta^{13}$C. Other records show only a moderate to weak cooling (Daley et al., 2010; Nesje et al., 2001; Muschitiello et al., 2013). The general picture that emerges from the data is that of westward decreasing cooling with increased distance from Eastern Europe/Western Asia. We did not find winter temperature proxies for SW Europe and the Middle East to fulfill our selection criteria; the majority of the proxies from this region are usually sensitive to precipitation amount changes.

Cold winters in Europe are associated with either blocking conditions over Central Europe or westward expansion of the high pressure cell – the Siberian High – centered over Asia (Cohen et al., 2001; Rîmbu et al., 2014; Ionita et al., 2018). In the Northern Hemisphere (NH), during the winter season, three semi–permanent and quasi–stationary systems prevail over the mid to high–latitudes: the Icelandic Low (over the Atlantic Ocean), the Aleutian Low (over the Pacific Ocean) and the Siberian High (SH). The SH is a semi–permanent anticyclone centered over Eurasia and is associated with cold and dense air masses in the NH and extreme cold winters over Europe and Asia (Cohen et al., 2001). The composite maps of the SH index and SLP and 10 m wind are shown in Fig. 2. As expected, in the case of positive SH index (HIGH years, Fig. 2a) an extensive area of strong and positive SLP anomalies prevail over the whole Eurasian landmass, with the highest anomalies over Siberia. The positive anomalies in Fig. 2 were found to be statistically significant at 5% level using a two–sample t-test. This SLP structure is associated with enhanced easterlies and advection of cold air towards Europe (blue background in Fig. 1a). For the years with a low index of the SH (Fig. 2b), negative SLP anomalies prevail over Siberia, while positive SLP anomalies are found over the central part of Europe. This kind of dipole–like structure in the SLP field associated with low SH years leads to the advection of warm air from the Atlantic Ocean basin towards the eastern part of Europe.

The robust association between the instrumental–based response of European/Asia temperatures to a strong SH (base map in Fig. 1) and the proxy–based reconstructions of winter air temperatures (blue dots in Fig. 1a) supports the hypothesis that a strengthened SH was active at the time of the 4.2 ka BP event (the possible mechanisms are described below). The

seasonality of the SH implies its onset in mid–autumn, likely linked to diabatic heating anomalies initiated by snow cover development in NE Siberia (Foster et al., 1983; Cohen et al., 2001). The cooling resulting from the expanding snow cover leads to anomalously high SLP in NE Asia, which in turn, results in more snowfall and further strengthening of the SLP anomaly. The rapidly developing high pressure and cold anomaly extends westwards, being limited towards north and east by the warm ocean SSTs (Cohen et al., 2001). The end result of an enhanced SH is a westward rolling high pressure system that also brings cold air, heavy snowfall and strong winds, both towards Europe and central Asia (Ding and Krishnamurti, 1987; Gong and Ho, 2002; Panagiotopoulos et al., 2005). The development of the SH also leads to strengthening of the subtropical jet stream over SE China (Panagiotopoulous et al., 2005), a characteristic feature of the East Asia Winter Monsoon (EAWM, Cheang, 1987) and instrumental data (Wu and Wang, 2002; Jhun and Lee, 2004) show that strengthening of the SH results in a stronger than average EAWM. Paleoclimate data from Asia further indicates the strengthening of the EAWM at 4.2 ka cal BP (e.g., Hao et al., 2017; Giosan et al., 2018), likely linked to stronger and more frequent outbreaks of cold air from the core of the SH. Similarly, paleoclimate records from the outer limits of the region impacted by the SH have documented significant increases in the strength of the local winds, frequently a local diagnostic signature of the 4.2 ka BP event. Various proxies in different sedimentary archives across West Asia have documented strong northerly winds at 4.2 ka cal BP: soil micromorphology at Tell Leilan (NE Syria, Weiss et al., 1993), detrital dolomite and calcite in Gulf of Oman (Cullen et al., 2000) and Red Sea (Arz et al., 2006) marine cores, high Ti counts in Lake Neor, Iranian plateau (Sharifi et al., 2015) and S/Ti ratios in Lake Kinneret, Israel (Vossel et al 2018), lake bed sediments in the UAE (Parker et al., 2006).

The strengthened EAWM and high windiness in SW Asia are consistent with the climatology of the SH, with strong clockwise flow of anomalously cold air from its center of action, located in north–central Asia (Fig. 2a). Paleoclimate records from Europe also document 4.2 ka BP–related increases in wind strength and or storminess, as at the raised bogs in SW Sweden (linked to cold temperatures and possible increased sea ice, Bjorck and Clemmensen, 2004), aeolian sand banks in coastal Denmark (Clemmensen et al., 2003; Goslin et al., 2018) and Gotland, Baltic Sea (Muschitello et al., 2013) (Fig. 3) where strong winter winds and high precipitation, the product of Baltic Sea moisture delivered by intense easterly winds indicate the reinforcement and westwards expansion of the Siberian High. These data suggest that a belt of strong winds extended around the core region of the SH, from East Asia through the West Asia and SE Europe up to the Baltic and North Seas (Fig. 3).

Summarizing the above information, at ca. 4.2 ka BP a cold temperature anomaly settled over most of Europe, from the Ural Mountains to the Atlantic Ocean, including Scandinavia and extending to the region south and east of the Caspian Sea, likely the result of a deeper than average Siberian High. Further, anomalously high SLP over this region resulted in the strengthening of winter winds in east, south and southwestern Asia and eastern and northeastern Europe, linked to clockwise and outward movement of cold air from the core of the SH–impacted region.

**3.2 Inconsistent winter precipitation patterns across Europe and southwest Asia**

Data from winter precipitation records at the time of the 4.2 ka BP event suggest a far more complex image of precipitation
distribution across our study area (Fig. 1b), as compared with the simpler temperature distribution dipole (Fig. 1a). The SE
Mediterranean and the wider Middle East were dry (Bini et al., 2018), with some of the droughts occurring rather abruptly
(Cheng et al., 2015; Sharifi et al., 2015). In the wider Mediterranean Basin, winter drought was also recorded in S Greece
(Finné et al., 2017), north–central Italy (Drysdale et al., 2006; Regattieri et al., 2014; Isola et al., 2018), N Algeria (Ruan et
al., 2016) and central Spain (Smith et al., 2016), all records pointing towards an abrupt onset and a ca. 150–200 years
duration. On this background of generalized drought in the Mediterranean, in two regions an increase in winter precipitation
amounts was registered (Fig. 1b), most notably in NW Africa and SW Europe (Walczak et al., 2015; Wassenburg et al.,
2016; Zielhofer et al., 2017) and the Central Balkans and Carpathian Mountains (Zanchetta et al., 2012; Panait et al., 2017;
Perşoiu et al., 2017). Multiple records and different proxies (speleothem and lake sediment $\delta^{18}$O, peatbog $\delta^{13}$C, cave ice d-
excess and growth rate) indicate similarly wet conditions, clearly underscoring the wet nature of climate at that time in these
two regions. The high winter precipitation amounts registered by records in the Balkan Peninsula and the Carpathian
Mountains (Fig. 1b) occurred during periods of intense cold (Fig. 1a). Winter precipitation in the Carpathian Mountains is
the result of either eastward advection of wet air masses of Atlantic origin, or precipitation from northward travelling
Mediterranean cyclones encountering the NE winds induced by a strong SH. The $\delta^{18}$O and d-excess records from Scărişoara
Ice Cave (Perşoiu et al., 2017) indicate that at 4.3 ka cal BP, late autumn through early winters were cold and the moisture
source was shifted to an area of high evaporation (as indicated by the high d-excess values). Modern monitoring of stable
isotopes in precipitation in the region (Drăguşin et al., 2017; Ersek et al., 2018; Bădăluţă et al., in press) indicates that high
d-excess values occur when the source of moisture is either the Eastern Mediterranean Sea or the Black Sea. A Black Sea
source for the moisture leading to high precipitation in the Carpathian Mountains is consistent with the information of
prevailing northeasterly winds at 4.2 ka BP (see section 3.1. above), but it would not fully explain the possibly wet
conditions on the Adriatic Coast at 4.3 ka cal BP (Fig. 1b, Zanchetta et al., 2012), where high winter precipitation is the
result of moisture originating in the Adriatic Sea (Ulbrich et al., 2012). We note however, that the Adriatic coast could also
have been dry at 4.2 ka BP, as suggested by a spike in the carbonate $\delta^{18}$O record of Shkodra Lake (Zanchetta et al., 2012).
Interestingly, the response of present–day climatic conditions in Europe to a stronger than usual Siberian High is of low SLP
in the Central Mediterranean Sea (centered on Italy, Fig. 2a), which in turns results in enhanced cyclogenesis in the area.
Thus, in the case of strong SH conditions at 4.2 ka BP, enhanced cyclogenesis would have resulted in more frequent NW
movement of moisture–bearing weather systems, further leading to higher than average precipitation on the Adriatic Coast
and the Carpathian Mountains (Fig. 1b). Apart from the high d-excess in the Scărişoara Ice Cave record (Perşoiu et al., 2017)
at 4.3 ka BP, indicative of Mediterranean moisture, the ice accumulation rate also reached a maximum at that time,
suggesting high precipitation amounts and early onset of freezing conditions in the cave, both favorable for the rapid growth
of ice (Perşoiu et al., 2011).
Apart from the SW Europe, the Balkans and the Carpathian Mts., high precipitation at 4.2 ka BP in Europe was also
registered in a lake at the foothills of the Alps (Cartier et al., 2019) and in Gotland, the Baltic Sea (Muschitielo et al., 2013).
In the Alps, high flooding activity at 4.2 ka BP was linked to increased autumn precipitation (Cartier et al., 2019), while in
the Baltic, high winter precipitation is consistent with strong easterly winds picking–up local moisture form the Baltic Sea
(Muschitielo et al., 2013, as well as the discussion in 3.1 above).
The winter precipitation record in Europe and the Middle East can now be summarized as follows (Fig. 1b):
1) regionally significant dry conditions occurred during winter in the Middle East, southern Europe (Italy and Greece),
northern Africa, as well as on a band stretching from the Atlantic Ocean, through the north European plains, towards eastern
Europe, including Scandinavia;
2) regionally significant wet conditions occurred during winter around the Gibraltar Straight (northern Morocco and
southern Spain) and in the northern Balkan Peninsula (including the Carpathian Mountains).
The distribution of precipitation minima and maxima on the western (Atlantic) side of Europe is similar to that occurring
during the negative phase of the North Atlantic Oscillation (NAO), one of the main modes of climate variability in Europe
(Hurrell et al., 2013), mainly active during winter. The NAO is defined as the difference in atmospheric pressure between the
Icelandic Low and the Azores High. A below average difference between the two pressure system (negative NAO, or NAO-)
results in weaker than usual and southwards deflected westerly winds, carrying more moisture towards southern Europe. As
precipitation amounts are negatively correlated with the NAO phase in the western Mediterranean (i.e., NAO- results in high
precipitation, Lionello et al., 2006), the reconstructed distribution of precipitation at 4.2 ka BP (Fig. 1b), partly supports the
hypothesis of prevailing NAO- conditions during the 4.2 ka BP event. Proxy–based reconstructions of the NAO index (Olsen
et al., 2012) indicate a brief negative mode at 4.2 ka cal BP, but contradictory evidence from speleothem and pollen data
from the Central Mediterranean region (e.g., Bini et al. (2018) and references therein) suggest that a combination of different
mechanisms (including NAO- conditions) could have been responsible for the winter climatic conditions at 4.2 ka BP in
Europe.
**3.3 The Siberian High in the global context at 4.2 ka**
The paleoclimate evidence we have compiled collectively suggests cold winter conditions in N Asia and Europe, likely
induced by cold air outbreaks from high pressure fields located over Siberia, conditions that in modern climates are
associated with a strong Siberian High. The sole reconstruction of the past behavior of the Siberian High is based on analysis
of the continental–sourced nssK$^+$ (non–seasalt potassium) in Greenland ice cores (Mayewski et al., 1994; O'Brien et al.,
1995). Meeker and Mayewski (2002) have shown that in years with high nssK$^+$ deposits in Greenland, the SLP over N Asia
in spring (indicator of the strength of the SH) is higher than average, thus providing a possible proxy for the strength of the
Siberian High. The reconstructed values for the strength of the SH (using the original data of Mayewski et al. (1997) on the
GICC05modelext timescale (Seierstad et al., 2014) shows a maximum at around 4.3 ka BP, in agreement within dating
uncertainties with paleoclimate data presented in Fig. 1.
Previous studies, based on instrumental, tree ring and ice core impurity content have shown a clear link between strong
SH and cold and dry climate in Europe (Meeker and Mayewski, 2002, D'Arrigo et al., 2005), and the close match between

the impact of the SH on temperature and precipitation amounts and the reconstructed climate (Fig. 1) suggest that at 4.2 ka BP a stronger than usual SH lead to cooling in Asia and Europe, disruption of the westerlies and drought in the Middle East (Fig. 3). The possible causes of this chain of events remains, however, elusive. Some possible forcings behind climate changes do not appear abruptly at 4.2 ka BP. Orbital forcing resulted in low winter insolation in the N Hemisphere and comparably high, but decreasing, summer insulation, while radiative forcing was going through a remarkably long state of stable, albeit high, values (Steinhilber et al., 2009). Volcanic and greenhouse forcing were both low and stable at 4.2 ka BP, with no abrupt changes (e.g., Wanner et al., 2011). The high contrast between summer and winter insolation would have resulted in a weak polar vortex (Orme et al., 2017) and thus more meridional polar vortex and associated southward displaced storm tracks in the Atlantic. The same meridional displaced polar vortex could have lead to cold air advection to N Asia and early onset of the winter, with earlier formation of the snow cover.

The early presence and persistence of snow in NE Asia is one of the most important triggers of a strong SH (Cohen et al., 2001; Wu and Wang, 2002). The causes and mechanisms by which snow accumulates in early winter in NE Asia are elusive, with possible causes being a positive feedback from the NAO, with NAO- conditions in late winter/early spring leading to early beginning of snow accumulation in the following winter and subsequent onset of a strong SH (Bojariu and Gimeno, 2003). The NAO index (Olsen et al., 2012) shows a continuous change from NAO+ to NAO- conditions after 4.5 ka BP, with a distinct negative excursion at 4.2 ka BP. A weak/negative NAO would have resulted in low wind stress and associated enhancement of the salinity stratification in the North Atlantic, initiating the slowdown of the Atlantic Meridional Overturning Circulation (AMOC, Yang et al., 2016). Thornalley et al. (2009) have documented a rapid and abrupt reduction in salinity at 4.2 ka BP that could have triggered the weakening of the AMOC. Reduced strength of the AMOC could have further led to southward expansion of sea ice and thus further decrease in salinity and weakening of the AMOC (Yang et al., 2016). Further, negative NAO conditions are also linked to a weakening of the subpolar gyre (Eden and Jung, 2001; Häkkinen and Rhines, 2004) and thereby reduced contribution of freshwater to the AMOC and further cooling in the Nordic Seas. Similarly, weak NAO conditions result in stronger northeastern winds and increase in the strength of the East Greenland current and associated sea ice export, further leading to the weakening of the thermohaline circulation (Orme et al., 2018) and subsequent cooling of the North Atlantic, as seen in both paleodata and models (e.g., Rîmbu et al., 2003; Renssen et al., 2005; Berner et al., 2008; Sejrup et al., 2016; Orme et al., 2018). In turn, these conditions led to reduced SLP around Iceland and reinforcement of the negative NAO.

The above inferences suggest that at ca. 4.2 ka BP, orbital and solar forcing led to a chain of atmospheric changes, transmitted and amplified by ocean circulation, which caused abrupt cold and dry climatic conditions in northern Eurasia. These atmospheric changes included the weakening of the polar vortex and southward advection of cold air over N Asia. The enhanced meridional transport generated earlier and more persistent autumn snow cover. In turn, this led to the onset of a stronger than usual Siberian High that lowered Eurasian surface temperatures with strong outbreaks of cold and dry northerly winds in a belt stretching from eastern Asia through portions of west Asia and central and northern Europe. The above average SLP associated with the strengthened SH resulted in the blocking of the moisture–bearing westerlies in Europe.

Megadrought across the Mediterranean and west Asia may also have been enhanced by the weak and southward–displaced
Atlantic storm track that resulted from lower than average NAO conditions. The conditions associated with a weak polar
vortex strengthened sea ice towards the Nordic Seas, further contributing to the weakening of the thermohaline circulation
and reduction in the strength of the NAO and of the westerlies.
**Conclusions**
We have gathered records of changes in winter temperature, precipitation amount and associated climatic conditions in the
wider Eurasian region during the 4.2 ka BP event. The data show that 4200 years ago cold winter temperature anomalies
dominated western Asia and most of Europe. The strength of winter winds in eastern and southern Asia was strongly
enhanced, while those in western Europe weakened. Regionally significant droughts settled over the Middle East, southern
and northern Europe and western Asia, while locally significant increases in precipitation were reconstructed in the Balkan
Peninsula, the Carpathian Mountains, around the Baltic Sea and in NW Africa and southern Spain.
We propose a multi–causal hypothesis of partially mutual reinforcing vectors and mechanisms to explain the regionally
coherent north Eurasian and adjacent region 4.2 ka BP phenomena. Thus, we hypothesize that before and at 4.2 ka BP, the
orbitally–induced high insolation gradient between summer and winter in the high–latitudes of the Northern Hemisphere led
to a weakening of the polar vortex, resulting in a meandering jet that promoted an early onset of winter season in NE Siberia.
In turn, this resulted in decreasing temperatures and an early and stronger Siberian High that expanded south and westwards,
bringing cold and dry conditions across Eurasia. The same circulation pattern lead to more sea ice export in the North
Atlantic and weakening of the subpolar gyre resulting in the slowdown of the thermohaline circulation and decrease of sea
level pressure around Iceland, thus possibly leading to a shift towards a negative phase of the North Atlantic Oscillation. In
turn, these changes resulted in weaker and southward displaced westerly winds across Europe. However, the high pressure
systems in Europe effectively blocked these weakened westerlies, causing reduced winter precipitation and drought
conditions across the eastern Mediterranean and western Asia. Clockwise circulation around the Asia–centered high pressure
field induced strong northerly winds in southern and western Asia and in eastern Europe. Further, the strong thermal pressure
gradient between central and northern Asia and the Indian and Pacific oceans determined the strengthening of the East Asian
and Indian Winter Monsoons. However, given the drought in the source regions of the winter monsoon, these strengthened
winds did not result in increased moisture advection. Nevertheless, several regions experienced a slight increase in winter
precipitation due to strong winds picking up moisture from local sources (NW Africa, N Balkan Peninsula and the
Carpathian Mountains, the Baltic region).
In the context of the above data and description, we suggest that, in the extra tropical regions of Eurasia, the 4.2 ka BP
event was a century–scale boreal winter phenomenon. While not the subject of our study, we note that a clear antiphase
behavior of the winter and summer monsoons have been evidenced (Kang et al., 2018), suggesting that at the times when
parts of Asia and Europe were experiencing winter droughts related to strong, dry, winter monsoons, SE Asia was

experiencing similar summer droughts, resulting from failed and/or reduced monsoons. Whether these were caused by the same orbitally induced changes and/or teleconnections transmitted via the weakened AMOC are questions to be investigated within future proxy–based and modeling studies. Especially important would be winter precipitation records from Western Asia and Eastern Europe, as well winter temperature records from southern Europe and the wider Middle East, where such data are scarce. Further, most of the winter records are of low resolution and/or with poor chronological control, such that improvements in these fields are required to further test our hypothesis.

**Data availability.** All data in this study has been obtained from the cited references.

**Author contributions.** AP designed the hypothesis, AP and HW collected, reviewed and analyzed the paleoclimate data, AP and MI discussed the climatology of the SH, AP synthesized the evidences and wrote the text with input from HW and MI.

**Competing interests.** The authors declare that they have no conflict of interest

**Acknowledgments.** The Scărișoara ice core analyses in Romania were partially supported by UEFISCDI Romania through grants no. PN-III-P1-1.1-TE-2016-2210 and PNII-RU-TE-2014-4-1993 awarded to AP, ELAC2014/DCC-0178/FP7, and from contract 18PFE/16.10.2018 funded by Ministry of Research and Innovation in Romania within Program 1 - Development of national research and development system, Subprogram 1.2 - Institutional Performance -RDI excellence funding projects. AP further acknowledges support from SP-PANA-W1010. *Associazione Italiana per lo studio del Quaternario* and the organizers of the "4.2 ka BP Event: An International Workshop" (Pisa, Italy) financially supported AP to attend the workshop where some of the ideas presented here were born. MI was funded by the Helmholtz Climate Initiative REKLIM and by the Polar Regions and Coasts in the Changing Earth System (PACES) program of the AWI. We thank the editor, Giovanni Zanchetta, and two anonymous referents for their comments.

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

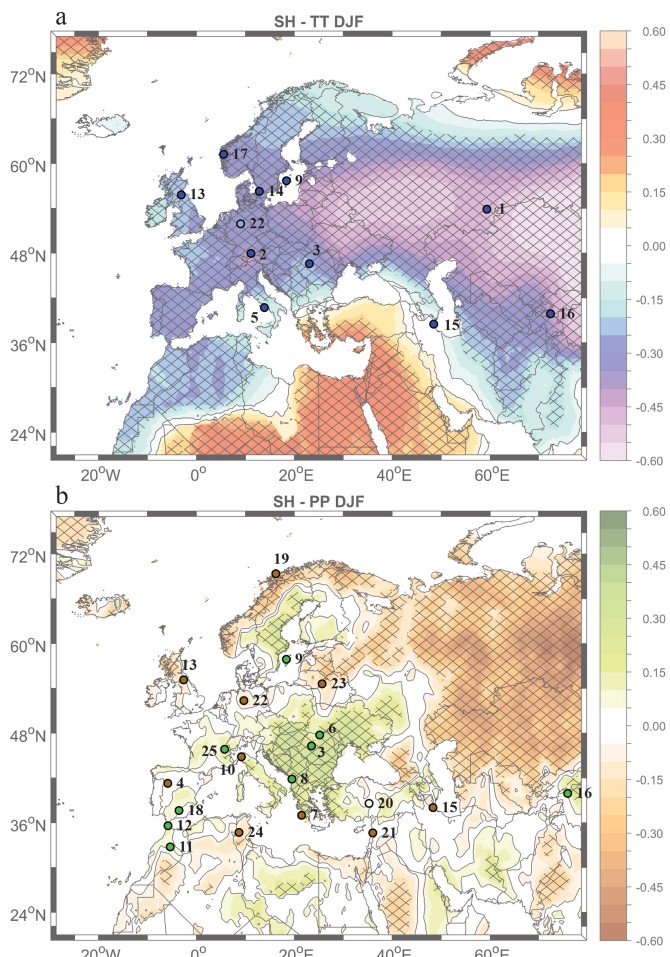


**Figure 1:** Climatic conditions at 4.2 ka cal BP in Europe and Western Asia. The background map in (**a**) shows the correlation between the
winter SH index and the winter mean temperature (December-January-February, DJF), with blue (red) shading indicating cold (warm)
winters. The dots indicate winter climatic conditions at 4.2 ka cal BP. The background map in (**b**) shows the correlation between the
winter SH index and winter precipitation (DJF), with green (brown) indicating wet (dry) winters. Green (brown) dots in (**b**) indicate
wet (dry) conditions at 4.2 ka cal BP. The hatched areas in (**a**) and (**b**) indicate correlations significant at 95% significance level based
on a Student t-test. The numbers in (**a**) and (**b**) correspond to the archives listed in Table 1.

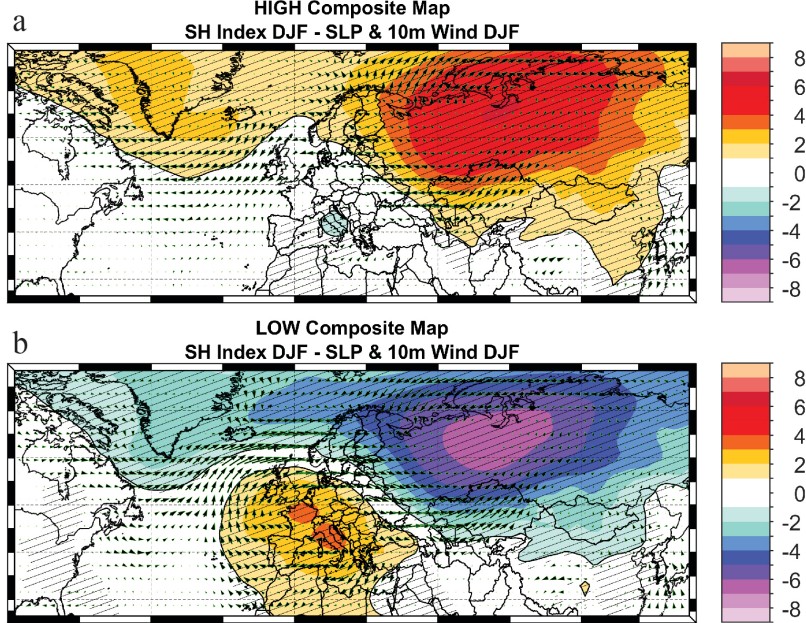

**Figure 2:** The composite map of the winter (DJF) sea level pressure (SLP) and wind at 10 m for the years when the SH index > 1 standard deviation (**a**) and the composite map of the winter (DJF) sea level pressure (SLP) and wind at 10 m for the years when the SH index < - 1 standard deviation (**b**). The hatching highlights significant SLP anomalies at a confidence level of 95% based on a Student t-test. The SLP units are in hectopascals (hPa).

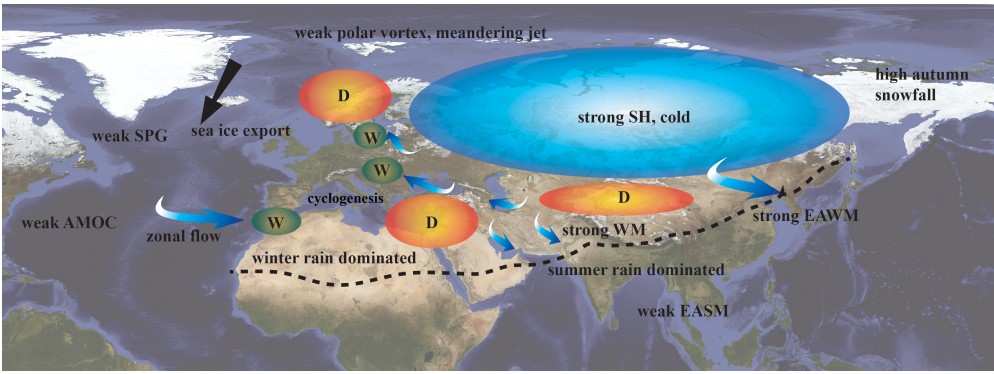

**Figure 3:** Inferred winter climatic conditions between ~ 4.3 ka and 3.9 ka cal BP. The position of the polar vortex is only indicative. The base map shows the Earth's surface conditions during November (Reto Stöckli, NASA Earth Observatory).

| No | Name | Proxy | Indicator of | Proxy interpretation | yrs/sample | Reference |
|---|---|---|---|---|---|---|
| 1 | Kinderlinskaya | Speleothem $\delta^{18}O$ | $T_W$ | Low values = cold | 12.5 | Baker et al., 2017 |
| 2 | Spannagel Cave | Speleothem $\delta^{18}O$ | $T_W$ | High values = cold, NAO- | 5 yrs | Fohlmeister et al., 2013 |
| 3 | Scărişoara | Ice $\delta^{18}O$ | $T_W$ | Low values = cold | 10 | Perşoiu et al., 2017 |
| | | d-excess | $M_{source}$ | High values = Mediterranean PP | | |
| 4 | Asiul Cave | Speleothem $\delta^{18}O$ | $PP_w$ | Low values = high precipitation | 1-28 | Smith et al., 2016 |
| 5 | Gulf of Gaeta | G. ruber $\delta^{18}O$ | $PP_w$ | Low values = high water inflow | 55 | Di Rita et al., 2018 |
| | | Globigerinoides % | $T_w$ | High values = cold | | |
| 6 | Tăul Muced | Sphagnum $\delta^{13}C$ | $PP_w$ | High values = wet | 8 | Panait et al., 2017 |
| 7 | Mavri Trypa | Speleothem $\delta^{18}O$ | $PP_w$ | High values = dry | 5 | Finne et al., 2017 |
| 8 | Shkodra Lake | Carbonate $\delta^{18}O$ | $PP_w$ | High values = low precipitation | <50 | Zanchetta et al., 2012 |
| 9 | Lake Bjarstrask | Gastropode $\delta^{18}O+\delta^{13}C$ | $PP_w$ | High values = wet winters | 80 | Muschitiello et al., 2013 |
| 10 | Buca della Renella | Speleothem $\delta^{18}O$ | $PP_w$ | High values = dry | 37 | Drysdale et al., 2006 |
| 11 | Sidi Ali Lake | $CaCO_3$ content | $PP_w$ | Low values = high lake level | 40 | Zielhofer et al., 2017 |
| | | Ostracod $\delta^{18}O$ | $PP_w$ | Low values = high % of pp | 130 | |
| 12 | Grotte de Piste | Speleothem $\delta^{18}O$ | $PP_w$ | Low values = wet | 15 | Wassenburg et al., 2016 |
| 13 | Walton Moss | Sphagnum $\delta^{18}O$ | $T_w$ | Low values = cold | 80 | Daley et al., 2010 |
| | | Multiproxy | $PP_w$ | Low values = dry | | |
| 14 | Hyltemossen | Minerogenic content | Wind | Low values = weak winds | | Bjorck&Clemmensen, 2004 |
| 15 | Neor Lake | Al, Zr, Ti, Si content | Dryness | High values = dry | 3.6 | Sharifi et al., 2015 |
| 16 | Uluu Cave | Speleothem $\delta^{13}C$ | $PP_w$ | Low values = wet/cold | 38 | Wolff et al., 2017 |
| 17 | Jostedalsbreen | Grain size variations | $PP_w$ | Low values = dry winters | 21 | Nesje et al., 2001 |
| 18 | Refugio | Stalagmite density | $PP_w$ | Low values = dry winters | 5 | Walczak et al., 2015 |
| 19 | Nattmasvatn | Minerogenic input | $PP_w$ | Low values = dry | - | Janbu et al., 2011 |
| 20 | Nar Golu Lake | Diatom $\delta^{18}O$ | $PP_w$ | Low values=more winter rainfall | 5 | Dean et al., 2018 |
| 21 | Jeita Cave | Speleothem $\delta^{18}O$ | $PP_w$ | High values = dry | 7 | Cheng et al., 2015 |
| 22 | Bunker Cave | Speleothem Mg/Ca | $PP_w$ | High values = dry | - | Wassenburg et al., 2016 |
| 23 | Nuudsaku Lake | Carbonate $\delta^{18}O$ | $PP_w$ | High values = dry winters | 13 | Stansell et al., 2017 |
| 24 | Gueldaman Cave | Speleothem $\delta^{18}O$ | $PP_w$ | High values = dry | - | Ruan et al., 2016 |
| 25 | Lake Petit | Detrital input | $PP_w$ | High values = wet | - | Cartier et al., 2019 |

**Table 1.** List of proxies used and their interpretation. Numbers in the first column corresponds to numbers in Fig. 1.