# Peer review of "Atmospheric blocking induced by the strengthened Siberian High led to drying in west Asia during the 4.2 ka BP event – a hypothesis"

_Climate of the Past, 2018_

## Referee Comment (RC1) · Anonymous Referee #1 · 12 Jan 2019

Perşoiu et al. present a very clearly written hypothesis attempting to explain the underlying climate dynamics that accounted for the widespread 4.2ka event. While much is known about the character and extent of the 4.2ka event, what is lacking is understanding of the causes, so the hypothesis presented here is very useful. Thinking about the seasonality of climate takes this study further than previous syntheses of this event. It is well thought through and argued. Figure 3 presents a useful summary of their hypothesis. The hypothesis itself is plausible. Future work – producing better records from more sites – is now required to help test this hypothesis. Good, thorough methodology for choosing which sites to include in your study.

I have very few suggestions for changes. One is that Dean et al. 2017 (actually 2018) is cited in the table but not in the reference list. Also, I know this is a synthesis and is focussed on the climatology, however because the hypothesis comes from the proxy data, I wonder if you could plot at least some of the proxy data on a summary graph. This would help with your argument and help readers to assess for themselves what the proxy data show. The authors could also go into more detail on what type of palaeo records, and from where, are required to properly test this hypothesis.

In summary, this paper is very well written and presents a useful and plausible hypothesis so I recommend it for publication with some changes.

In answer to the specific questions asked of reviewers for CP https://www.climate-of-the-past.net/peer_review/review_criteria.html : 1. Yes 2. Yes 3. Yes, to a certain extent 4. Yes 5. Yes 6. N/a 7. Yes 8. Yes 9. Yes 10. Yes 11. Yes 12. N/a 13. No 14. Yes 15. Yes

---

## Referee Comment (RC2) · Anonymous Referee #2 · 6 Mar 2019

The study focuses on the boreal winter season in Eurasia during the 4.2 cal BP event, where climate variability is a function of the spatio-temporal dynamics of the westerly winds. The authors present a multi-proxy reconstruction of winter climate conditions in Europe, west Asia and northern Africa between 4.3 and 3.8 ka BP. The authors hypothesise that in the extratropical Northern Hemisphere, the 4.2 ka BP event was caused by the strengthening and expansion of the Siberian High, which effectively blocked the moisture-carrying westerlies from reaching W Asia, and enhanced outbreaks of cold and dry winds in that region. The authors further hypothesise that in extratropical regions of Eurasia the 4.2 ka BP event was a century-scale winter phenomenon, whereas in the monsoon-dominated regions it may have been a feature of summer

climate conditions.

Generally, the introduction is well written, the aim and hypothesis of the current study are clear.

Methodological approach: the method chapter clearly documents the methodological steps and criteria.

Results and discussion: the authors create a convincing concept for the large-scale atmospheric conditions around 4.2 cal ka BP. I like the deduction of probable negative NAO-like conditions at 4.2 cal ka BP although the Olsen record (Olsen et al. 2012) does not indicate a clear negative NAO-like stage at that time.

Overall, the manuscript is well organised and I recommend publication in Climate of the Past with minor revisions.

Detailed comments: Line 15: delete "using"

Line 22: What do you mean exactly with "antiphase behaviour"?

Line 34: This is not clear for the Western Mediterranean. There are also indications for wet conditions at 4.2 cal ka BP in the W Mediterranean (e.g. Fletcher et al. 2013).

Line 90: "Our analysis has shown that the results are not sensitive to the exact threshold value used for our composite analysis". What does it mean? Explain the consequences for your data interpretation.

Fletcher, W.J., Debret, M., Sanchez Goñi, M.F.: Mid-Holocene emergence of a low-frequency millennial oscillation in western Mediterranean climate: implications for past dynamics of the North Atlantic atmospheric westerlies. The Holocene 23, 153-166, 2013.

Olsen, J., Anderson, N. J., and Knudsen, M. F.: Variability of the North Atlantic Oscillation over the past 5,200 years, Nature Geosci, 5, 808-812, 2012.

---

## Author Comment (AC1) · 20 Mar 2019

We thank the reviewer for the comments and suggestions. Our response is detailed below (in black), below the corresponding comments (in blue)

Perşoiu et al. present a very clearly written hypothesis attempting to explain the underlying climate dynamics that accounted for the widespread 4.2ka event. While much is known about the character and extent of the 4.2ka event, what is lacking is under- standing of the causes, so the hypothesis presented here is very useful. Thinking about the seasonality of climate takes this study further than previous syntheses of this event. It is well thought through and argued. Figure 3 presents a useful summary of their hypothesis. The hypothesis itself is plausible. Future work – producing better records from more sites – is now required to help test this hypothesis. Good, thorough methodology for choosing which sites to include in your study.

Thank you for the appreciations.

I have very few suggestions for changes. One is that Dean et al. 2017 (actually 2018) is cited in the table but not in the reference list. Also, I know this is a synthesis and is focused on the climatology, however because the hypothesis comes from the proxy data, I wonder if you could plot at least some of the proxy data on a summary graph. This would help with your argument and help readers to assess for themselves what the proxy data show. The authors could also go into more detail on what type of palaeo records, and from where, are required to properly test this hypothesis.

In summary, this paper is very well written and presents a useful and plausible hypothesis so I recommend it for publication with some changes.

We have corrected the reference to Dean et al. (2017).

Winter records are scarce, and, except for few, generally with low resolution. The records listed in Table 1 indicate the conditions at 4.2 ka cal BP as plotted in Figure 1, but not all of them show a clear-cut drop in the recorded values. We do not support cherry-picking "selected" records to be plotted in a figure. Further, all of the papers in the special issue (The 4.2 ka BP climatic event) have such plots (especially the review papers by Kaniewsky et al. and Bini et al.). We have added a line at the end of the text (in conjunction with the suggestion for future studies, see below) to further support our decision.

„...more detail on what type of palaeo records, and from where, are required to properly test this hypothesis…"

This is a very useful suggestion. We have added a sentence at the end of the "Conclusions", where we highlight several locations for future research to test (support or invalidate) our hypothesis. The text of the paragraph is appended below.

"Especially important would be winter precipitation records from Western Asia and Eastern Europe, as well winter temperature records from southern Europe and the wider Middle East, where such data are scarce. Further, most of the winter records are of low resolution and/or with poor chronological control, such that improvements in these fields are required to further test our hypothesis."

---

## Author Comment (AC2) · 20 Mar 2019

We thank the reviewer for the comments and suggestions. Our response is detailed below (in black), below the corresponding comments (in blue)

The study focuses on the boreal winter season in Eurasia during the 4.2 cal BP event, where climate variability is a function of the spatio-temporal dynamics of the westerly winds. The authors present a multi-proxy reconstruction of winter climate conditions in Europe, west Asia and northern Africa between 4.3 and 3.8 ka BP. The authors hypothesise that in the extratropical Northern Hemisphere, the 4.2 ka BP event was caused by the strengthening and expansion of the Siberian High, which effectively blocked the moisture-carrying westerlies from reaching W Asia, and enhanced outbreaks of cold and dry winds in that region. The authors further hypothesise that in extratropical regions of Eurasia the 4.2 ka BP event was a century-scale winter phenomenon, whereas in the monsoon-dominated regions it may have been a feature of summer climate conditions.
Generally, the introduction is well written, the aim and hypothesis of the current study are clear.
Methodological approach: the method chapter clearly documents the methodological steps and criteria.
Results and discussion: the authors create a convincing concept for the large-scale atmospheric conditions around 4.2 cal ka BP. I like the deduction of probable negative NAO-like conditions at 4.2 cal ka BP although the Olsen record (Olsen et al. 2012) does not indicate a clear negative NAO-like stage at that time.
Overall, the manuscript is well organised and I recommend publication in Climate of the Past with minor revisions.
Thank you for the appreciations.
On NAO- conditions at 4.2 ka cal BP: In Olsen et al. (2012), a short-lived excursion towards negative NAO conditions at ~4250 cal BP is shown in figure 3, interrupting a very long period of NAO+ conditions.

Detailed comments: Line 15: delete "using"
Done.

Line 22: What do you mean exactly with "antiphase behaviour"?
We mean a strengthening of the winter monsoon occurring synchronously with a weakening of the summer (Kang et al., 2018). Strong and dry winter monsoon would result in drought, as would do weak summer monsoons. It is true that without such an explication the question of "antiphase behavior" remains. We have thus deleted the word "behavior" from the abstract and kept in the main text (Lines 285-287) where we have clarified this "behavior" and included the above-mentioned reference.

Line 34: This is not clear for the Western Mediterranean. There are also indications for wet conditions at 4.2 cal ka BP in the W Mediterranean (e.g. Fletcher et al. 2013).
True. We have used the wetter conditions in the western Mediterranean region to support our hypothesis of prevailing NAO- conditions. We have changed the text to reflect this, as follows: "These studies have defined the spatial extent and variability of the event. Megadrought developed abruptly at ca. 4.2 ka cal BP across North America, Andean South America, the Mediterranean basin from Spain to Turkey (except for several records from N Morocco and S Spain which indicate wetter conditions), Iran, India, Tibet, and north China and Australia […]."

Line 90: "Our analysis has shown that the results are not sensitive to the exact threshold value used for our composite analysis". What does it mean? Explain the consequences for your data interpretation.
In the manuscript we choose for the composite map analysis the years when the SH index was greater (HIGH) and lower (LOW) than a value of one standard deviation. This is an arbitrary threshold (the state of the art threshold in climatology), but if we vary the threshold (e.g., 0.5 standard deviation or 1.5 standard deviation) the spatial structure and the significance of the composite maps remains the same. This was meant by the statement: "Our analysis has shown that the results are not sensitive to the exact threshold value used for our composite analysis". We have changed the text to make this clear, as follows:
Our analysis has shown that the results are not sensitive to the exact threshold value used for our composite analysis (i.e., varying the standard deviation between 0.5 and 1.5).

Additional reference:
Kang, S., Wang, X., Roberts, H. M., Duller, G. A. T., Cheng, P., Lu, Y., and An, Z.: Late Holocene anti-phase change in the East Asian summer and winter monsoons, Quat. Sci. Rev., 188, 28-36, 2018.

---

## Author Response (AR1)

Comments to the Author:
The work is an excellent contribution to the special volume. I have very few comments. Mostly technical, but some to clarify the text.
**We have made the suggested modifications to the text, see our responses below and the attached text with the changes highlighted.**

Line 187: in the special issue Cartier et al. 2018 indicate a possible different climate reconstruction. Consider to revise this point.
**We have modified the text considering the new record of Cartier et al. (2019). The text now reads:**
"Apart from the SW Europe, the Balkans and the Carpathian Mts., high precipitation at 4.2 ka BP in Europe was also registered in a lake at the foothills of the Alps (Cartier et al., 2019) and in Gotland, the Baltic Sea (Muschitielo et al., 2013). **In the Alps, high flooding activity at 4.2 ka BP was linked to increased autumn precipitation (Cartier et al., 2019),** while in the Baltic, high winter precipitation is consistent with strong easterly winds picking–up local moisture form the Baltic Sea (Muschitielo et al., 2013, as well as the discussion in 3.1 above)."

Lines 197-207: note that clear aridity indicated by pollen and speleothem records from central Mediterranean are not coherent with NAO-.
**We have amended the text, it reads:**
"Proxy–based reconstructions of the NAO index (Olsen et al., 2012) indicate a brief negative mode at 4.2 ka cal BP, but contradictory evidence from speleothem and pollen data from the Central Mediterranean region (e.g., Bini et al. (2018) and references therein) suggest that a combination of different mechanisms (including NAO- conditions) could have been responsible for the winter climatic conditions at 4.2 ka BP in Europe."

Some corrections are necessary for Table 1

8: in their manuscript Zanchetta et al. 2012 consider the indication of 4.2 event (even if slightly younger) as drier for a spike with higerh delta18O-values. Consider this also in the text.
**The text has been modified as suggested. It now reads:**
"A Black Sea source for the moisture leading to high precipitation in the Carpathian Mountains is consistent with the information of prevailing northeasterly winds at 4.2 ka BP (see section 3.1. above), but it would not fully explain the possibly wet conditions on the Adriatic Coast at 4.3 ka cal BP (Fig. 1b, Zanchetta et al., 2012), where high winter precipitation is the result of moisture originating in the Adriatic Sea (Ulbrich et al., 2012). **We note however, that the Adriatic coast could also have been dry at 4.2 ka BP, as suggested by a spike in the carbonate δ18O record of Shkodra Lake (Zanchetta et al., 2012)."**

10 Buca della (not dela) Renella: high values drier conditions
**Done.**

12 Grotte de Piste: in the list your wrote high values = dry. It should be lower values = wetter, as correctly reported within the text.
**Corrected.**

After these small corrections the manuscript can be accepted.

My compliments to the authors!
**Thank you.**